# Target Localization and Grasping of NAO Robot Based on YOLOv8 Network and Monocular Ranging

Yingrui Jin [1,*], Zhaoyuan Shi [2], Xinlong Xu [2], Guang Wu [1], Hengyi Li [3] and Shengjun Wen [1,*]

1   School of Zhongyuan-Petersburg Aviation, Zhongyuan University of Technology, Zhengzhou 450007, China
2   School of Electronic and Information Engineering, Zhongyuan University of Technology, Zhengzhou 450007, China; 2021106208@zut.edu.cn (Z.S.)
3   Department of Electronic and Computer Engineering, Ritsumeikan University, Kusatsu 525-0058, Shiga, Japan
*   Correspondence: yingrui.jin@zut.edu.cn (Y.J.); wsj@zut.edu.cn (S.W.)

**Abstract:** As a typical visual positioning system, monocular ranging is widely used in various fields. However, when the distance increases, there is a greater error. YOLOv8 network has the advantages of fast recognition speed and high accuracy. This paper proposes a method by combining YOLOv8 network recognition with a monocular ranging method to achieve target localization and grasping for the NAO robots. By establishing a visual distance error compensation model and applying it to correct the estimation results of the monocular distance measurement model, the accuracy of the NAO robot's long-distance monocular visual positioning is improved. Additionally, a grasping control strategy based on pose interpolation is proposed. Throughout, the proposed method's advantage in measurement accuracy was confirmed via experiments, and the grasping strategy has been implemented to accurately grasp the target object.

**Keywords:** intelligent robots; object recognition; YOLOv8; visual odometry; error compensation; pose estimation





## 1. Introduction

With the rapid development of robotics technology, robots have been widely used in various fields such as transportation, welding, and assembly [1]. However, the precise positioning and grasping of robots are key technologies and prerequisites for them to carry out a variety of tasks. Zhang L. et al. proposed a robotic grasping method that uses the deep learning method YOLOv3 and the auxiliary signs to obtain the target location [2]. The method can control the movement of the robot and plan the grasping trajectory based on visual feedback information. However, the detection accuracy of this method is relatively low since target detection in the robot's image is complicated by a complex background and target occlusion. Huang M. et al. proposed a multi-category SAR image object detection model based on YOLOv5s to address the issues caused by complex scenes [3]. Tan L. et al. adopted the hollow convolution to resample the feature image to improve the feature extraction and target detection performance [4]. The improved YOLOv4 algorithm has been adopted by numerous studies to facilitate target detection in robotic vision, aiming to enhance detection accuracy [5,6]. The improved algorithm has improved both detection speed and accuracy. Sun Y. et al. constructed the error compensation model based on Gaussian process regression (GPR), effectively improving the accuracy of positioning and grasping for large-sized objects [7]. This study focuses on the target localization and grasping of the NAO robot [8,9]. The target object is recognized through YOLOv8 network training [10]. Then, a monocular ranging model is established for the NAO robot to achieve the initial positioning of the target. Then, we establish a visual distance error compensation model to improve the monocular range. Furthermore, we utilize multi-point measurement compensation technology to estimate the target's position and pose and ultimately achieve grasping the target.

The main contributions include: (1) Combining the YOLOv8 network with the Nao robot's monocular ranging to realize the target recognition and localization and improving the issue of targets being unable to be accurately segmented in complex environments; (2) Due to the issue of greater errors with longer distances in the monocular distance measurement model, we propose a visual distance error compensation model using Gaussian process regression to improve the Nao robot's distance ranging error within 2 cm; (3) The multi-point measurement compensation technology is proposed to estimate the target's position and pose, and ultimately achieve grasping the target.

This paper is organized as follows: In Section 2, relevant target recognition and Localization technology is reviewed. In Section 3, the visual distance error compensation model is established to improve the long-distance monocular visual positioning accuracy of the Nao robot. In Section 4, a grasp control strategy based on pose interpolation is proposed to realize the pose estimation and smooth grasping and further verify the results of target recognition and localization. The experiment and results analysis are given in Section 5. Finally, the discussions are drawn in Section 6.

## 2. Target Recognition and Localization Technology

Target recognition based on traditional color segmentation has high requirements for the environment in which the target object is situated. In contrast, the YOLOv8 network has the advantages of fast speed, high accuracy, and strong scalability. Through training, the latter can extract feature points from the target to achieve target recognition [11]. The Nao robot operates using a single camera. Hence, this study employs the monocular vision localization techniques [12,13]. A monocular vision system is a typical visual positioning system [14]. But, most of the present research faces the problem that the farther the distance is, the greater the error is, and the attitude of the target object is not considered. First, the position coordinates of the target center under the image coordinate system are obtained through target detection using the YOLOv8 network. Then, the relationship between the location coordinates and image coordinates was determined using the monocular vision positioning model. Finally, it is possible to obtain the location coordinates of the target under the NAO robot coordinate system and to acquire the pose of the target object by measuring the endpoint and the center point of the target object, thereby ensuring that the NAO robot can accurately grasp the object.

The principle of monocular ranging based on the YOLOv8 algorithm is shown in Figure 1. The system mainly consists of three components: target detection, internal and external parameter acquisition, and monocular ranging. In the target detection phase, the detection frame is mainly obtained by the YOLOv8. Then, the internal and external parameters of the camera are obtained through the calibration. Finally, according to the principle of similar triangle ranging, the center point of the frame output in the target detection phase is used as the mapping point and combined with the internal and external parameters of the camera to achieve target positioning.

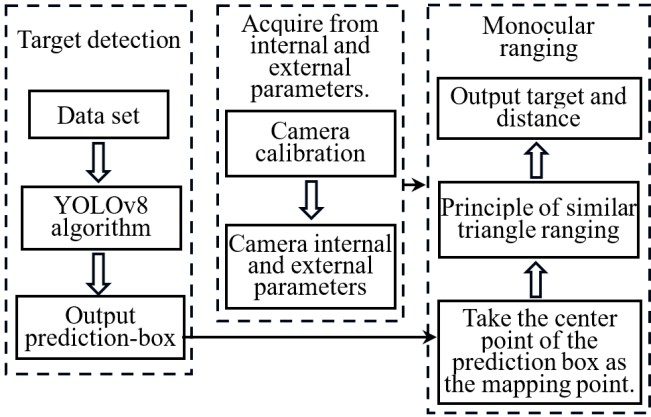

**Figure 1.** Schematic diagram of monocular ranging based on YOLOv8.

### 2.1. Target Recognition Based on YOLOv8 Network

YOLOv8 is a deep neural network architecture used for target detection tasks, as shown in Figure 2. Compared to YOLOv5 and YOLOv7, it provides a new SOTA model and a YOLACT-based instance segmentation model [11]. It is built as a unified framework for training object detection, instance segmentation, and image classification models. And the model can run on both CPU and GPU. The network consists of four main components: the input, the backbone, the loss calculation, and the regression branch. The input layer primarily focuses on data enhancement techniques, such as Mosaic [11], to carry out adaptive detection frame calculation and filling of grayscale values. The backbone network is responsible for feature extraction. The loss calculation process comprises two branches, namely classification and regression, to achieve accurate and efficient target detection [15–17].

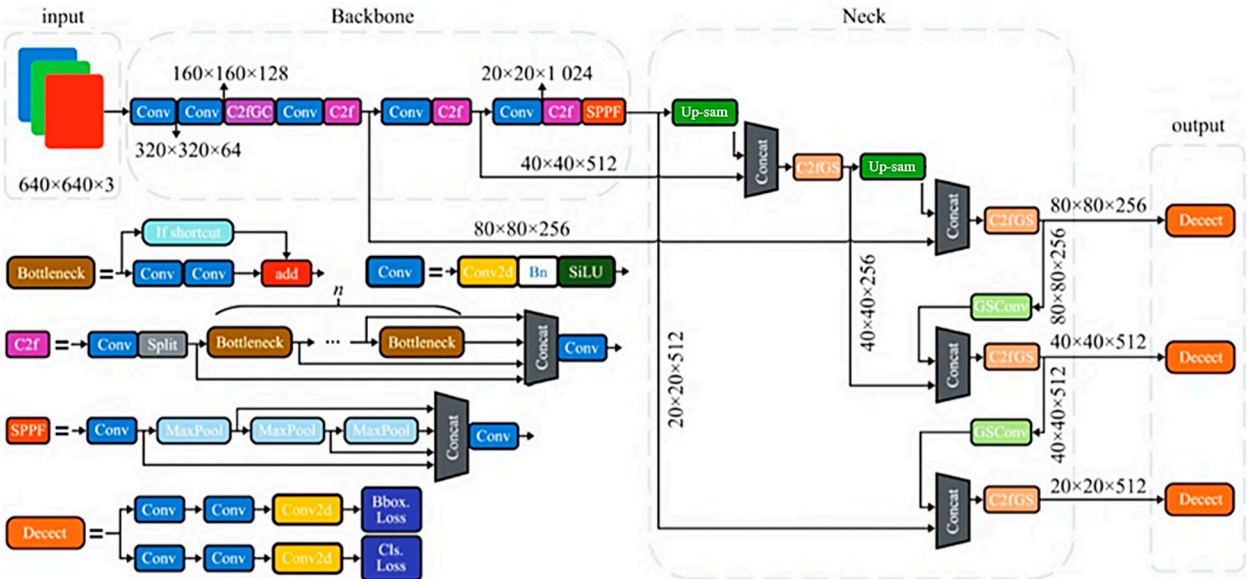

**Figure 2.** The structure of YOLOv8 network.

### 2.2. Modeling of Monocular Ranging

Based on the NAO robot, a monocular ranging model is employed, utilizing the pinhole perspective principle as depicted in Figure 3. The relationship between the camera coordinate system, and the image coordinate system $X - Y$ in the camera imaging model is represented below. With the optical center position $Oc$ of the camera lens as the origin point, a camera coordinate system $X_C - Y_C - Z_C$ is established parallel to the $X - Y$ coordinate plane. The $Z_C$ axis is perpendicular to the $X - Y$ coordinate plane. The line segment $O_C O_1$ represents the camera's focal length. The point $M$ of coordinates $(X_C, Y_C, Z_C)$, corresponds to the point $m$ in the image coordinate system, with coordinates $(X, Y)$. The relationship between image coordinates and actual spatial coordinates is depicted by Equation (1).

$$\begin{bmatrix} X \\ Y \\ 1 \end{bmatrix} = \frac{1}{Z_c} \begin{bmatrix} f & 0 & 0 & 0 \\ 0 & f & 0 & 0 \\ 0 & 0 & 1 & 0 \end{bmatrix} \begin{bmatrix} X_c \\ Y_c \\ Z_c \\ 1 \end{bmatrix} \tag{1}$$

where $f$ is the camera focal length.

Hence, after obtaining the coordinates of the center point and endpoint of the target image, the image coordinates can be converted into actual spatial coordinates. Subsequently, the pose of the target object in the robot coordinate system can be determined.

To determine the position of the target point in the image coordinate system, the center point $(u_0, v_0)$ of the image pixel is taken as the origin of the image coordinate system. The transformation relationship is depicted in Equation (2).

$$\begin{cases} x = (u - u_0)d_x \\ y = (v - v_0)d_y \end{cases} \tag{2}$$

where $dx$ and $dy$ represent the size of each pixel, and $u$ and $v$ correspond to the pixel coordinates of the target point.

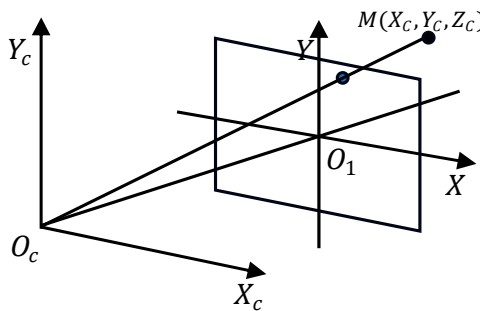

**Figure 3.** The pinhole imaging model.

Figure 4 shows the monocular ranging model established for the NAO robot. The robot is positioned at the origin $O_W$ within the coordinate system $O_W X_W Y_W Z_W$. Point $O$ serves as the camera position, and $xO_1y$ represents the image coordinate system. The endpoints $Q_1$, $Q_2$ of the target rod correspond to $q_1$, $q_2$ in the image coordinate system, respectively. Taking point $Q_1$ as an example, based on the principles of triangle similarity, the corresponding relationships of various angles can be obtained as shown in Equations (3)–(6).

$$f_y = \frac{f}{d_y} \tag{3}$$

$$\beta_1 = \alpha + \gamma_1 \tag{4}$$

$$\gamma_1 = \arctan\left(\frac{y}{f_y}\right) \tag{5}$$

$$X_W = \frac{H}{tan\beta_1} \tag{6}$$

Among them, $\alpha$ denotes the pitch angle of the NAO robot's head, $H$ represents the height of the camera above the ground, $f$ is the camera focal length, and $f_y$ denotes the component of $f$ in the image coordinate system. The value of $f$ has been calibrated to be 470 pixels.

By utilizing Equations (2)–(6), the X-coordinate $P_{X1}$ of point $Q_1$ can be derived, as depicted in Equation (7).

$$P_{X1} = \frac{H}{\tan\left(\alpha + \arctan\left(\frac{v - v_0}{f_y}\right)\right)} \tag{7}$$

The monocular ranging model for the NAO robot can be simplified into a perspective view, as shown in Figure 5. There, $\theta_1$ represents the angle between point $Q_1$ and the principal optical axis in the horizontal direction. As a result, the distance between the target point and

the robot in the *Y*-axis direction can be obtained. This is formulated in Equations (8)–(10), where $\varphi$ denotes the angle of the NAO robot's head in the horizontal direction.

$$f_x = \frac{f}{dx} \tag{8}$$

$$\theta_1 = \arctan\left(\frac{x}{f_x}\right) \tag{9}$$

$$P_{Y1} = Y_1 = P_{X1} \times \tan(\theta_1 + \varphi) \tag{10}$$

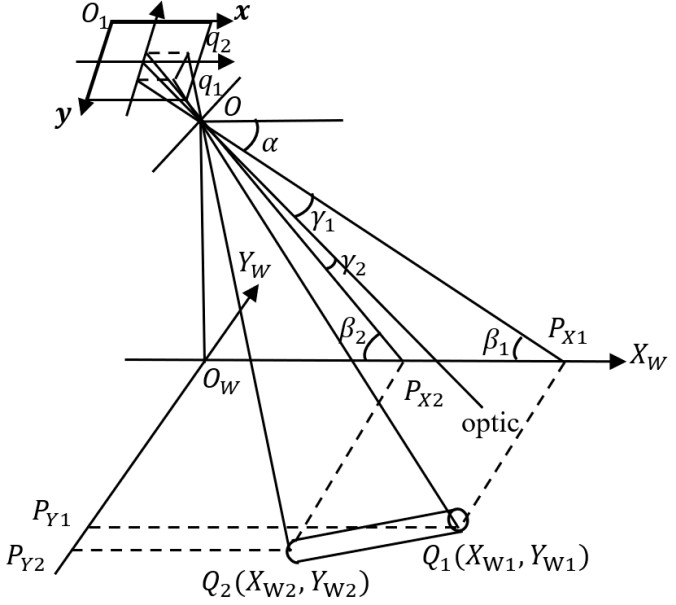

**Figure 4.** The monocular ranging model for the NAO robot.

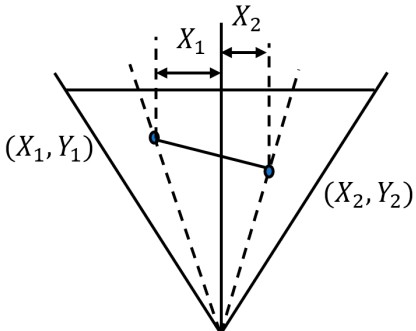

**Figure 5.** Vertical view of the monocular ranging model.

Similarly, it is possible to derive the position coordinates $(X_{W2}, Y_{W2})$ of point $Q_2$ under the robot's coordinate system.

After the NAO robot has measured an accurate distance, the estimation of the target object's pose can be achieved through a method of multiple measurements. By using the monocular ranging model established in Figure 4, range measurements are performed on the two endpoints of the target bar, thereby obtaining the coordinate values of $Q_1$ and $Q_2$, which are $(P_{X1}, P_{Y1})$ and $(P_{X2}, P_{Y2})$, respectively. Consequently, the deflection angle $\epsilon$ of the target rod on the $O_W X_W Y_W$ plane can be obtained, as demonstrated in Equation (11).

$$\epsilon = \arctan\left(\frac{P_{X1} - P_{X2}}{|P_{Y1}| + |P_{Y2}|}\right) \tag{11}$$

## 3. Modeling Visual Distance Error Compensation

### 3.1. Error Analysis

Based on the established monocular ranging model of the NAO robot, the distance in the *X*-axis direction of the robot's coordinate system is related to the $\gamma$ angle in a tangent function relationship, as shown in Figure 6. The further away, the smaller the $\gamma$ angle. This results in larger measurement errors for distances that are further away.

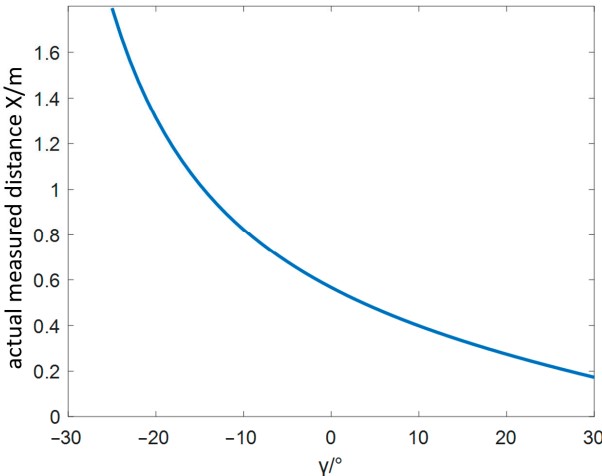

**Figure 6.** Relationship between the $\gamma$ angle and the measured distance.

Therefore, an error compensation model is established to reduce measurement errors when the target object is at a distance. The error term *k* has a relationship with the measured distance $d_m$ of the target rod. The relationship is depicted in Figure 7.

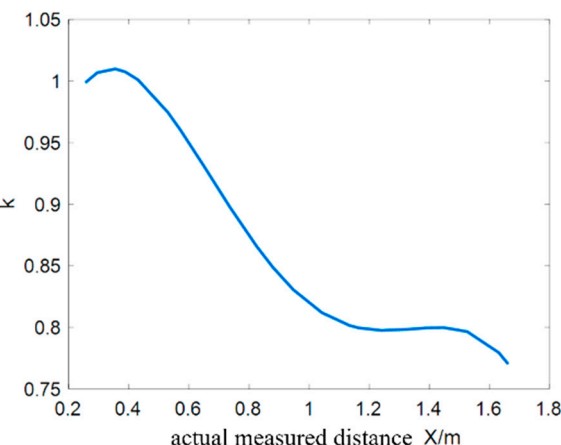

**Figure 7.** Relationship between the measured distance and error coefficient *k*.

### 3.2. Gaussian Process Regression Model

Gaussian process regression (GPR) is a method of regression analysis using Gaussian processes (GP). The Gaussian process is composed of a series of random variables following a normal distribution [18].

Suppose there is a learning sample $\left\{\vec{X}, y\right\}$ containing *n* independent observations, where $\vec{X} = \left\{\vec{x}_1, \vec{x}_2, \cdots, \vec{x}_n\right\}$ is an input table consisting of *n* input vectors, and $y = \{y_1, y_2, \cdots, y_n\}$ is an output table consisting of *n* corresponding one-dimensional outputs. So, the mean of GP, $\mu(\mathrm{x})$, and the variance $k(x, x')$ can be expressed by the Equations (12) and (13).

$$\mu(x) = E[f(x)] \tag{12}$$

$$k(x, x') = E[(f(x) - \mu(x))(f(x') - \mu(x'))] \tag{13}$$

Here, $x, x'$ represents any random variable in the sample, and $E$ is the mathematical expectations. Therefore, the GP can be defined as shown in Equation (14). In practical applications, the mean function $\mu(x)$ is zero through data preprocessing, so GP is as shown in Equation (15).

$$f(x) \sim GP\left[\mu(x), k(x, x')\right] \tag{14}$$

$$f(x) \sim GP\left[0, k(x, x')\right] \tag{15}$$

In regression problems, the output $y$ is also affected by noise $\varepsilon$, which follows the Gaussian distribution with a mean of zero and a variance of $\sigma_n^2$, so the output $y$ can be expressed as Equation (16).

$$y = f(x) + \varepsilon \tag{16}$$

Then, we can obtain the prior distribution of $y$, as Equation (17), and the joint prior distribution of $y$ and $f(x_*)$, as Equation (18). $x_*$ is the test input vector, $I$ is the n-dimensional identity matrix, and $K\left(\vec{X}, \vec{X}\right)$ is $n \times n$ order GP covariance matrix, $K\left(\vec{X}, x_*\right)$, is $n \times 1$ order covariance matrix between the input set $\vec{X}$ and the test input vector $x_*$, and $k(x_*, x_*)$ is $x_*$ its own covariance.

$$y \sim N\left(0, K\left(\vec{X}, \vec{X}\right) + \sigma_n^2 I\right) \tag{17}$$

$$\begin{bmatrix} y \\ f(x_*) \end{bmatrix} \sim N\left(0, \begin{bmatrix} K\left(\vec{X}, \vec{X}\right) + \sigma_n^2 I_n & K\left(\vec{X}, x_*\right) \\ K\left(x_*, \vec{X}\right) & k(x_*, x_*) \end{bmatrix}\right) \tag{18}$$

From this, the posterior distribution of the predicted value $f(x_*)$ can be obtained as shown in Equations (19)–(21). Predicted mean value $\overline{f}(x_*)$ is the output of the GPR model and the predicted value of the observed value $y$.

$$f(x_*) \mid \vec{X}, y, x_* \sim N\left[\overline{f}(x_*), \text{cov}(f(x_*))\right] \tag{19}$$

$$\overline{f}(x_*) = K(x_*, X)\left[k(X, X) + \sigma_n^2 I_n\right]^{-1} y \tag{20}$$

$$\text{cov}(f(x_*)) = k(x_*, x_*) - K(x_*, X)\left[K(X, X) + \sigma_n^2 I_n\right]^{-1} \times K(X, x_*) \tag{21}$$

This study selects the square exponential function [18] as the kernel function, as shown in Equation (22).

$$k(x_i, x_j \mid \theta) = \sigma_f^2 e^{\left[-\frac{1}{2} \frac{(x_i - x_j)^T (x_i - x_j)}{\sigma_l^2}\right]} \tag{22}$$

Among them, $\sigma_l$ and $\sigma_f$ are respectively the characteristic length scale and the signal standard deviation. The Gaussian model can be derived from prior data, as shown in Figure 8. Here, the values of $\sigma_l$ and $\sigma_f$ are 0.7344 and 1.0522 respectively.

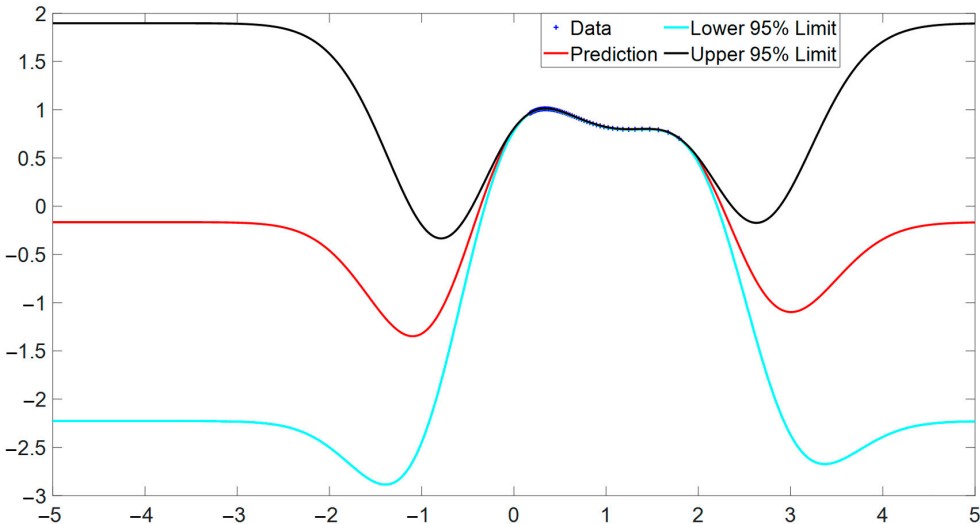

**Figure 8.** Gaussian process regression result.

The blue "+" is the true value of the function, and the red solid line is the result of Gaussian process regression. The upper and lower solid lines represent two samples of the Gaussian distribution, and the area between them is the confidence region. Both satisfy the mean value of the red solid line. The farther away from the red solid line, the larger the variance and the lower the confidence level.

By combining Equations (16) and (22), we can obtain the error compensation model Equation (23). And by combining Equations (7) and (10), the target coordinates after compensation are given by Equation (24).

$$y = \sigma_f^2 e^{\left[-\frac{1}{2}\frac{(x_i - x_j)^T (x_i - x_j)}{\sigma_l^2}\right]} + \varepsilon \tag{23}$$

$$\begin{cases} X_1 = P_{X1} \times y \\ Y_1 = P_{Y1} \end{cases} \tag{24}$$

## 4. Pose-Interpolated Grasping Control Strategy

To verify the results of target recognition and localization, we adopted a pose interpolation strategy to plan the trajectory of the Nao robot arm to achieve smooth grasping of the target.

### 4.1. Linear Path Interpolation

The path of the NAO robotic arm end effector from the start point to the endpoint follows a linear trajectory. Therefore, interpolation is applied to the straight path between the start and endpoints. Let the positional coordinates of workspace start and end points be denoted as $A = (x_a, y_a, z_a)$ and $B = (x_b, y_b, z_b)$, respectively. The distance between the start and end points is $L = \sqrt{(x_b - x_a)^2 + (y_b - y_a)^2 + (z_b - z_a)^2}$, A point $P_i$ on the line segment, $AB$ can be represented as $P_i = P_a + (P_b - P_a)S(t)/L$, $t \in [0, T]$, and its coordinates are denoted as Equation (25):

$$\begin{cases} x_i = x_a + \frac{S(t)(x_b - x_a)}{L} \\ y_i = y_a + \frac{S(t)(y_b - y_a)}{L} \\ z_i = z_a + \frac{S(t)(z_b - z_a)}{L} \end{cases} \tag{25}$$

The interpolation curves of displacement, velocity, and acceleration are depicted in Figure 9. The arm velocity and acceleration both become zero at the start and end of the movement, ensuring the stability of the robot arm throughout its motion.

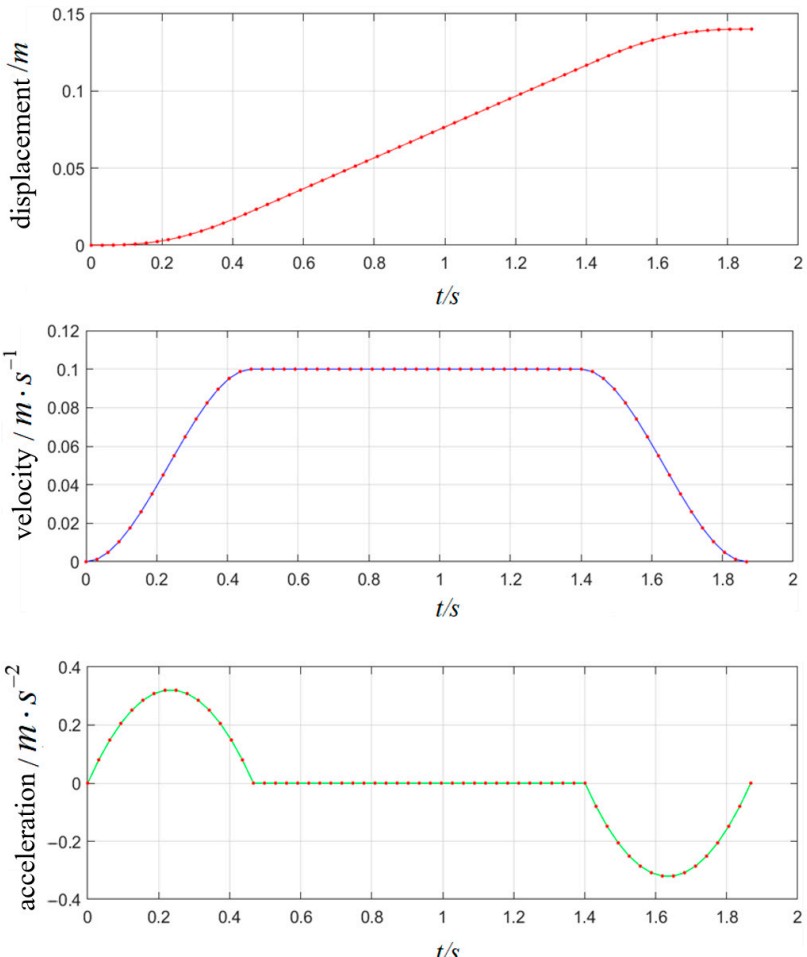

**Figure 9.** Interpolation curves for displacement, velocity, and acceleration.

Substituting the $S(t)$ from the acceleration-uniform-deceleration trajectory into the $x_i$, results in the arm's linear motion trajectory in space, as shown in Figure 10. It is evident that the points are densely packed at the ends of the straight line, while the middle portion is evenly distributed. This arrangement achieves the effect of acceleration-uniform-deceleration.

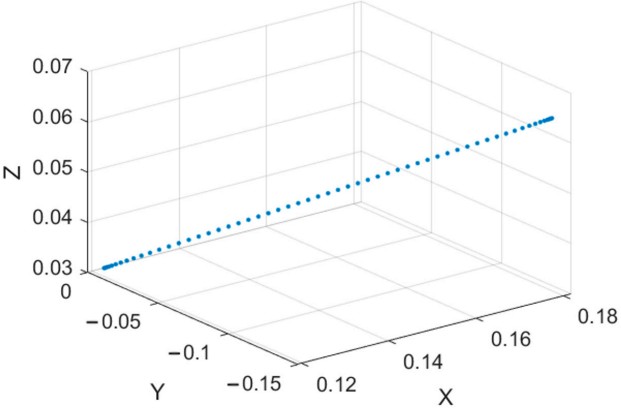

**Figure 10.** Linear motion interpolation diagram.

### 4.2. Position Interpolation

Due to the issue of excessive acceleration at the start and end points, we employ the fourth-order polynomial interpolation method for trajectory planning during the acceleration and deceleration phases.

The arm end displacement, velocity, and acceleration functions are expressed as $S(t)$, $V(t)$, and $A(t)$. The distance between the start and end points is denoted as $L$, and the velocity constant is represented as $V_m$, the time intervals for the three phases are represented as $t \in [0, T/4, 3T/4, T]$. $S(t)$, $V(t)$, and $A(t)$ of these three phases can be represented by the Equations (26)–(28) respectively. The acceleration phase $t \in [0, T/4]$, $S_1(t)$, $V_1(t)$, and $A_1(t)$ are

$$\begin{cases} S_1(t) = -\frac{V_m}{2t_1^3}t^4 + \frac{V_m}{t_1^2}t^3 \\ V_1(t) = -\frac{2V_m}{t_1^3}t^3 + \frac{3V_m}{t_1^2}t^2 \\ A_1(t) = -\frac{6V_m}{t_1^3}t^2 + \frac{6V_m}{t_1^2}t \end{cases} \tag{26}$$

The constant velocity phase $t \in [T/4, 3T/4]$, $S_2(t)$, $V_2(t)$, and $A_2(t)$ are

$$\begin{cases} S_2(t) = V_m t - V_m t_1/2 \\ V_2(t) = V_m \\ A_2(t) = 0 \end{cases} \tag{27}$$

The deceleration phase $t \in [3T/4, T]$, $S_3(t)$, $V_3(t)$, and $A_3(t)$ are

$$\begin{cases} S_3(t) = b_4 t^4 + b_3 t^3 + b_2 t^2 + b_1 t^1 + b_0 \\ V_3(t) = 4b_4 t^3 + 3b_3 t^2 + 2b_2 t + b_1 \\ A_3(t) = 12b_4 t^2 + 6b_3 t + 2b_2 \end{cases} \tag{28}$$

### 4.3. Pose Interpolation

There are two methods for solving the pose of the robotic arm: the Euler method and the quaternion method. However, the Euler method struggles with issues such as singularities and coupling of angular velocities. Therefore, the quaternion method is chosen to interpolate the arm posture of the NAO robot.

The relationship between the quaternion $q_t$ and arm end pose matrix $R$ is as shown in Equations (29)–(33), where $I$ is the identity matrix and $\omega$ is the anti-symmetric matrix.

$$\begin{cases} q_t = [q_0, q_1, q_2, q_3] = [q_0, q_x] \\ R = I + 2q_0\omega + 2\omega^2 \end{cases} \tag{29}$$

Convert the initial rotation matrix $R_b$ and the final rotation matrix $R_f$ into quaternions. And, then, the attitude angle $\theta$ is obtained.

$$\begin{cases} q_b = [b_0, b_1, b_2, b_3] \\ q_f = [f_0, f_1, f_2, f_3] \\ \theta = \cos^{-1}\left(q_b \bullet q_f\right) \end{cases} \tag{30}$$

At a certain moment $t$ within this period $T$, the rotation matrix is represented by the quaternion $q_t$ as follows:

$$q_t = x q_b + y q_f \tag{31}$$

where $x$, $y$ are real numbers and the attitude angle $\frac{t}{T}\theta$ between the initial quaternion $q_b$ and the quaternion $q_t$ at time $t$ is defined. The attitude angle $\left(1 - \frac{t}{T}\right)\theta$ between the quaternion $q_t$ at time t and the final quaternion $q_f$ is defined. Therefore, the quaternion pose interpolation matrix is

$$q_t = \frac{q_b \sin\left(\left(1 - \frac{t}{T}\right)\theta\right)}{\sin\theta} + \frac{q_f \sin\left(\frac{t}{T}\theta\right)}{\sin\theta} \tag{32}$$

By performing position interpolation, the displacement matrix *P* can be obtained. Similarly, through pose interpolation, the rotation matrix *R* can be derived. By combining the displacement matrix *P* and the rotation matrix *R*, the pose interpolation matrix is obtained. Subsequently, by solving the inverse kinematics of the pose interpolation matrix, the angle values of various joints during the NAO robot arm's motion process can be determined.

Conduct simulation experiments for arm trajectory planning by using MATLAB, taking two points coordinates as the starting and ending points of the arm movement, as illustrated in Equation (33).

$$\begin{cases} xyz\_\text{begin} = [0.1817, -0.1362, 0.0633] \\ xyz\_\text{fin} = [0.12, -0.01, 0.03] \end{cases} \tag{33}$$

Using these two points as the starting point and end point for trajectory planning, the corresponding pose interpolation matrix is substituted into the inverse kinematics equation, and arm motion simulation is performed using MATLAB to obtain the variation curve of the five joints from the ShoulderRoll to the WristYaw in the NAO robot's right arm, as shown in Figure 11. From the curves depicted in the graph, it is evident that the NAO robot's arm can move smoothly from the start point to the end point.

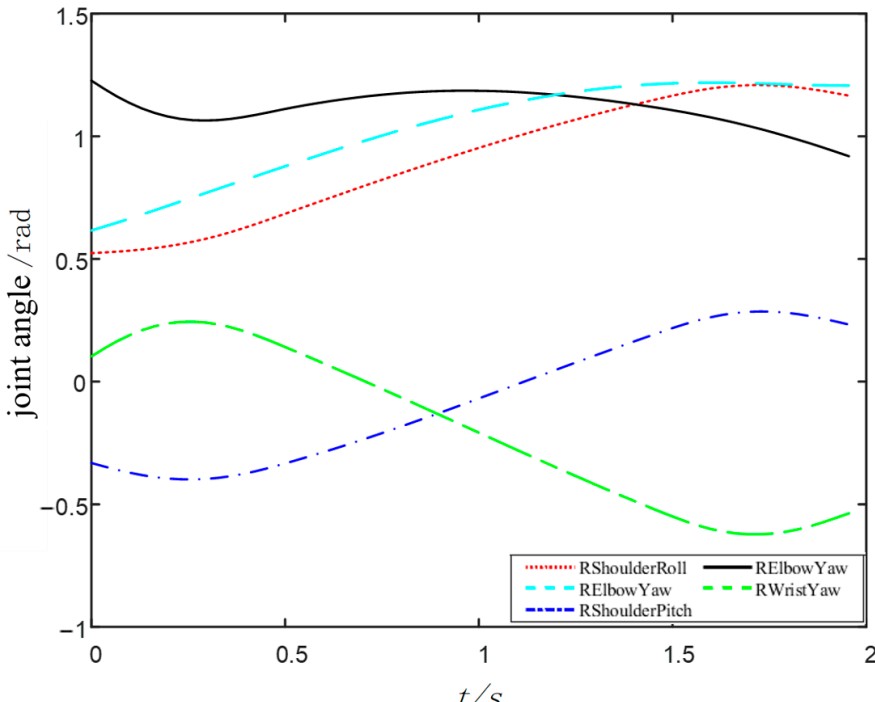

**Figure 11.** Joint angle motion curves.

## 5. Experiments and Results Analysis

### 5.1. The NAO Robot Platform

The NAO robot is a bipedal humanoid robot developed by Aldebaran Robotics. It has 25 degrees of freedom in its body. Its hardware equipment includes a CPU, ultrasonic sensors, gyroscopes, and infrared sensors.

The NAO robot's visual system incorporates two COMS cameras positioned on its forehead and mouth, which serve as a primary mode of perception for the robot in its surrounding environment. In the experiments discussed in this paper, the lower camera of the NAO robot will be utilized.

The right arm of the NAO robot has five degrees of freedom and is a serial robotic arm structure, with each joint connected by a link. According to the D-H method [19], the data shown in Table 1 can be obtained. On this basis, a link coordinate system for the right arm

of the NAO robot can be established, as shown in Figure 12. In this coordinate system, s, e, and w represent the shoulder, elbow, and wrist joints of the NAO robot, respectively. The dimensions of the links are obtained through the parameters of the NAO robot's right arm, where d3 = 0.9 cm and d5 = 10.855 cm.

**Table 1.** D-H parameters table of the right arm of NAO robot.

| Links | $\theta_i$ (°) | $d_i$/mm | $a_i$/mm | $\alpha_i$ (°) | Joint Range (°) |
|-------|----------------|----------|----------|----------------|-----------------|
| 1 | $\theta_1$ | 0 | 0 | 90 | −119.5~119.5 |
| 2 | $\theta_2$ (−90°) | 0 | 0 | −90 | −76~18 |
| 3 | $\theta_3$ | d3 | 0 | 90 | −119.5~119.5 |
| 4 | $\theta_4$ | 0 | 0 | −90 | 2~88.5 |
| 5 | $\theta_5$ (−90°) | d5 | 0 | 90 | 104.5~104.5 |
| 1 | $\theta_1$ | 0 | 0 | 90 | −119.5~119.5 |

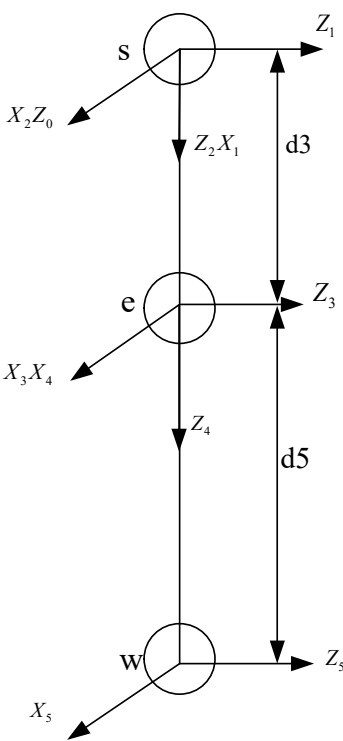

**Figure 12.** D-H model of NAO robot's right arm.

### 5.2. Object Detection Experiment

The experimental process begins with data collection using the NAO robot. A dataset is established and fed into the YOLOv8 network for training. The training of the YOLOv8 network is conducted on a Linux-based server.

In this experiment, the target object is a black rod, which is not specific. The NAO robot's bottom camera collected 100 images of the target rod at different angles, which were then processed through rotation and mirroring. Subsequently, the YOLOv8 network was trained for 800 rounds, with approximately 300 images per round. The original image captured by the NAO robot's camera is depicted in Figure 13a. The target bar is identified using the YOLOv8 network, resulting in a binary image of the target object, as shown in Figure 13b.

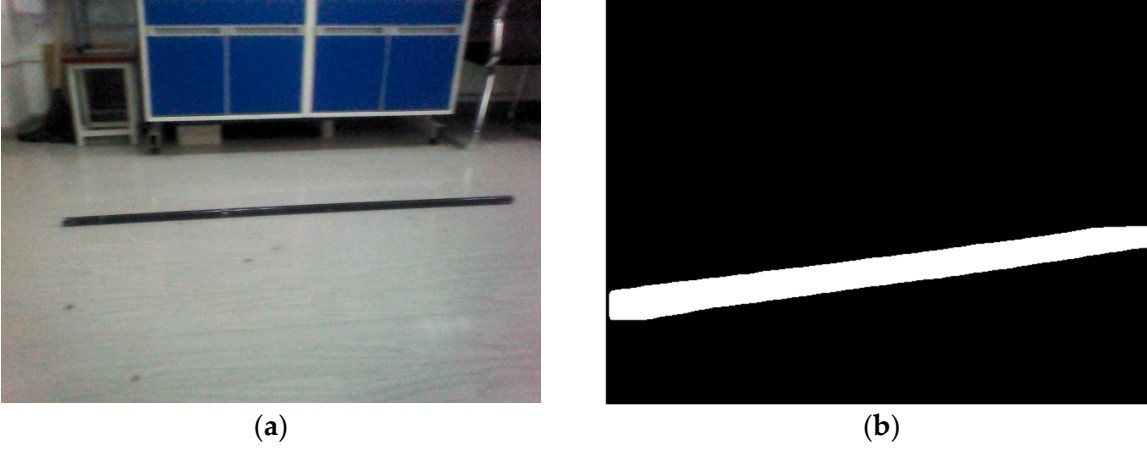

**Figure 13.** (**a**) Original image captured by NAO robot; (**b**) Target object identified by YOLOv8.

After obtaining the edge point information of the target object, as shown in Figure 14a,b, data processing is employed to extract the pixel coordinates of the object's center point and endpoints, and then the target is localized.

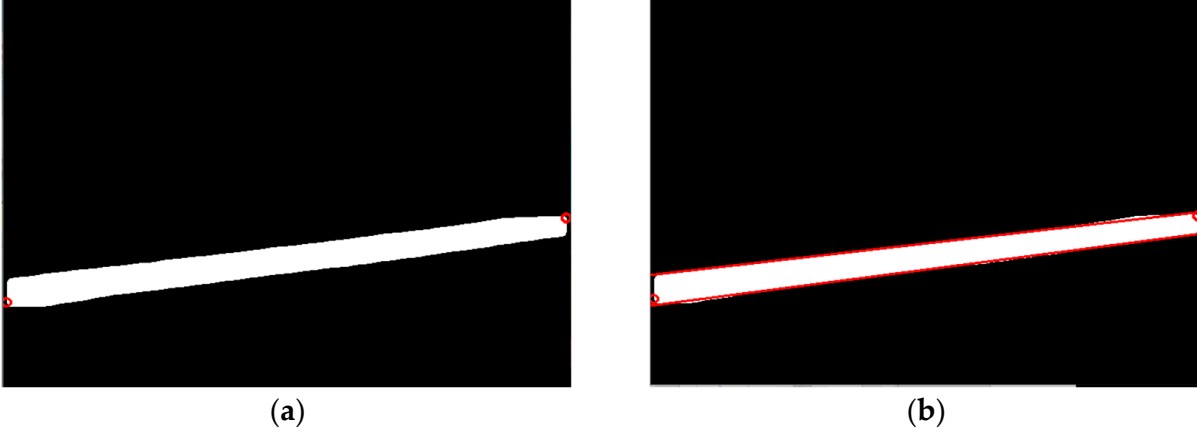

**Figure 14.** (**a**) Endpoints of the target object; (**b**) Edge of the target object.

The rod is positioned in front of the NAO robot at distances ranging from 0.25 m to 1.30 m, with intervals of 0.05 m. Multiple experiments are conducted at each position to calculate an average value. From Table 2, it can be observed that the farther the target is from the robot, the larger the error becomes. Beyond 60 cm, the distance error exceeds the requirements for the task.

**Table 2.** Actual and measured positions of the target before improvement.

| Actual Position (cm) | Measured Position (cm) | Actual Position (cm) | Measured Position (cm) |
|---|---|---|---|
| 25 | 25.50 | 80 | 94.71 |
| 30 | 29.49 | 85 | 104.26 |
| 35 | 35.45 | 90 | 113.50 |
| 40 | 38.83 | 95 | 116.43 |
| 45 | 43.84 | 100 | 123.99 |
| 50 | 52.94 | 105 | 133.08 |
| 55 | 57.17 | 110 | 139.06 |
| 60 | 64.80 | 115 | 144.73 |
| 65 | 73.69 | 120 | 152.62 |
| 70 | 82.55 | 125 | 163.13 |
| 75 | 87.94 | 130 | 166.29 |

To address the issue of significant measurement error when the target's position exceeds 60 cm, experiments were conducted using the improved monocular distance model with error compensation.

The target was placed in front of the NAO robot at distances ranging from 0.25 m to 1.30 m. From Table 3, it can be observed that the minimum error between the actual and measured positions is 0.13 cm, and the maximum error is 1.93 cm. Whether the target's position is before or after 0.6 m, the error does not exceed 0.02 m.

**Table 3.** Actual and measured positions of the target after error compensation.

| Actual Position (cm) | Measured Position (cm) | Actual Position (cm) | Measured Position (cm) |
|---|---|---|---|
| 25 | 25.47 | 80 | 78.67 |
| 30 | 29.69 | 85 | 84.64 |
| 35 | 35.80 | 90 | 90.96 |
| 40 | 39.12 | 95 | 93.10 |
| 45 | 43.08 | 100 | 98.80 |
| 50 | 51.60 | 105 | 106.25 |
| 55 | 54.89 | 110 | 111.18 |
| 60 | 60.37 | 115 | 115.75 |
| 65 | 66.13 | 120 | 121.56 |
| 70 | 71.46 | 125 | 127.16 |
| 75 | 74.64 | 130 | 128.07 |

As shown in Figure 15, the monocular distance measurement with the integrated error compensation model effectively reduces the distance error for positions that are farther away in the *X*-axis direction of the robot's coordinate system.

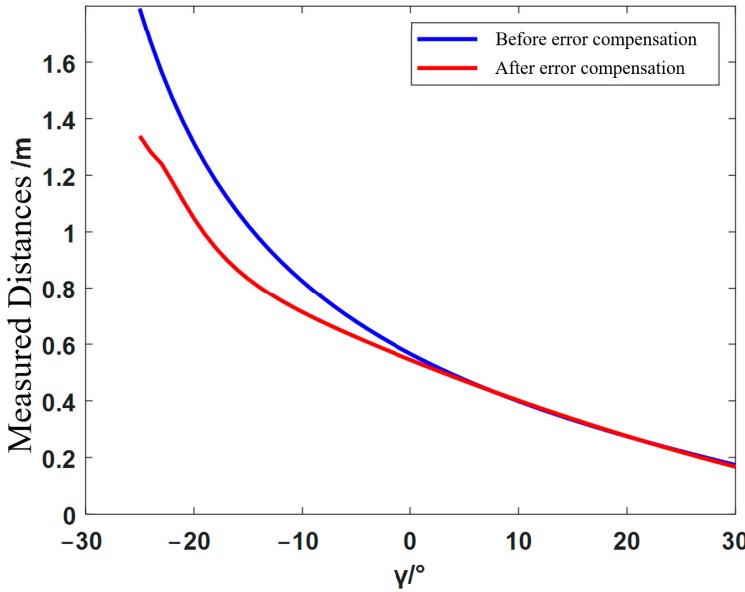

**Figure 15.** Comparison of measured distances before and after error compensation.

The rod was placed at 90 cm in the robot's X-direction, with distances of 0 cm, 20 cm, and 40 cm in the Y-direction. Each position underwent 10 tests, as shown in Table 4. The RMSEs of the three points are 0.644 cm, 0.574 cm, and 1.077 cm, respectively. It is evident that the NAO robot can accurately measure distances in the *Y*-axis direction, meeting the subsequent precision requirements.

**Table 4.** Actual and measured distances in the *Y*-axis direction after error compensation.

| Actual Distance (cm) | 0 | 20 | 40 |
|---|---|---|---|
| Index | | | |
| 1 | 0.8 | 20.4 | 41.2 |
| 2 | 0.7 | 20.9 | 40.4 |
| 3 | 0.8 | 19.5 | 40.4 |
| 4 | 0.6 | 20.4 | 41.5 |
| 5 | 0.8 | 20.3 | 40.4 |
| 6 | 0.6 | 19.7 | 41.7 |
| 7 | 0.4 | 20.2 | 41.4 |
| 8 | 0.5 | 20.9 | 41.5 |
| 9 | 0.6 | 20.8 | 39.6 |
| 10 | 0.5 | 20.5 | 40.4 |

After obtaining the position of the target rod, using the pixel coordinates of the two endpoints of the rod, the endpoint positions are calculated to determine the deviation angle of the rod. At a position of 60 cm in the robot's *X*-axis direction, measurements were taken for deviation angles $\alpha$ of 30°, 45°, and 60°. As shown in Table 5, the RMSEs are 0.820°, 0.904°, and 0.901°, respectively, so the NAO robot can effectively measure the deviation angle of the rod, providing a foundation for accurate grasping.

**Table 5.** Actual deviation angle vs. measured deviation angle.

| $\alpha/°$ | 30 | 45 | 60 |
|---|---|---|---|
| Index | | | |
| 1 | 30.48 | 45.66 | 60.85 |
| 2 | 30.76 | 45.69 | 59.36 |
| 3 | 30.53 | 45.93 | 58.82 |
| 4 | 31.22 | 45.87 | 59.56 |
| 5 | 29.87 | 46.05 | 60.59 |
| 6 | 30.82 | 45.92 | 60.89 |
| 7 | 30.63 | 46.15 | 61.35 |
| 8 | 29.08 | 45.56 | 61.09 |
| 9 | 29.35 | 44.58 | 60.53 |
| 10 | 31.34 | 46.37 | 59.02 |

*5.3. Object Grasping Experiment*

Due to the low friction between the ground and the feet of the NAO robot, it can experience slipping during walking, especially over longer distances. To mitigate this issue, a method involving measuring, short-distance walking, adjustment, and then measuring again. This approach ensures that the NAO robot can walk to the vicinity of the target rod with the correct orientation. Subsequently, adjust its crouching posture using the choreograph software. This ensures that the target rod is within the NAO robot's workspace. The internal API can obtain the position of its end effector. By combining this information with the known coordinates of the target's center point, the robot can accurately grasp the target at its center position. This process is illustrated in Figure 16.

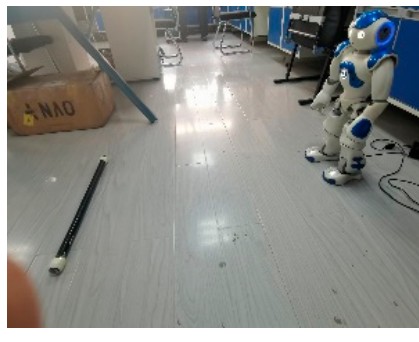 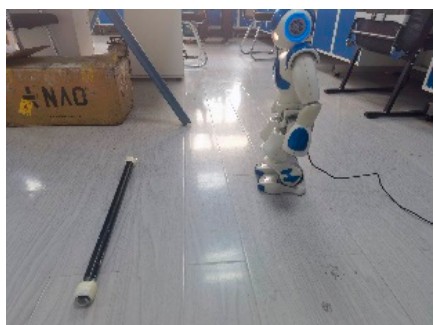

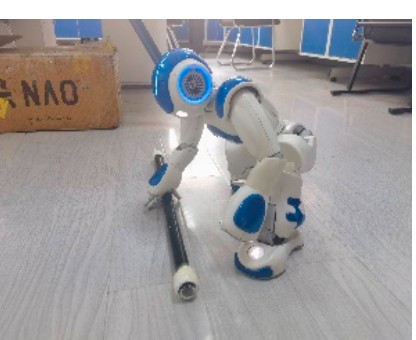 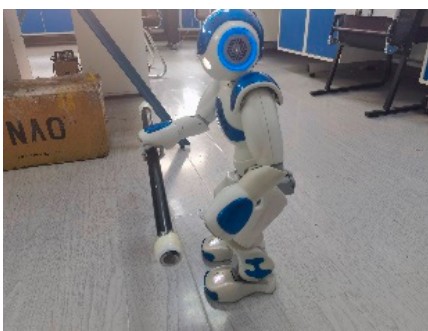

**Figure 16.** NAO robot grasping process.

## 6. Discussion

This paper combines YOLOv8 network recognition with monocular ranging methods to recognize and locate the target object. NAO robot acquires the pose information of the target object through its own monocular vision sensor, builds a visual distance error compensation model based on monocular ranging to compensate for distance errors, then moves near the target and grasps the target object by adjusting its attitude.

Based on experimental data and results, it is observed that the visual distance error compensation to the monocular ranging model effectively can improve the accuracy of the NAO robot's distance measurement. The error between the actual position and measurement position is controlled within 2 cm. This accuracy has been improved compared to other research results, such as the 5% error rate achieved by Wang Z. et al. through single-camera distance measurement [20]. Furthermore, by utilizing pose interpolation techniques, the pose of the finger is adjusted to align with the target at a constant level. The experimental results show that the rotation angle error is controlled within $2°$. These results indicate that the NAO robot can precisely estimate the target distance and pose and then facilitate precise walking and posture adjustments to ensure accurate object grasping. As previously mentioned, the accuracy of target recognition and long-distance localization in most present research has been improved.

**Author Contributions:** Conceptualization, Y.J. and S.W.; methodology, S.W.; software, Z.S.; validation, X.X. and G.W.; investigation, Y.J.; writing—original draft preparation, Z.S.; writing—review and editing, Y.J. and S.W.; project administration, H.L. All authors have read and agreed to the published version of the manuscript.

**Funding:** This work was supported by the National Natural Science Foundation of China (Grant No. 62073297), the Natural Science Foundation of Henan Province (Grant No. 222300420595), and the Henan Science and Technology research project (Grant No. 222102520024, No. 222102210019, No. 232102221035).

**Data Availability Statement:** Not applicable.

**Conflicts of Interest:** The authors declare no conflict of interest.

## Notations and Abbreviations

SOTA (State-Of-The-Art): Describe the model that achieves the current optimal effect on a certain task in machine learning; YOLACT (You Only Look At CoefficienTs): Real-time Instance Segmentation.

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
