# Peer review of "Target Localization and Grasping of NAO Robot Based on YOLOv8 Network and Monocular Ranging"

_electronics, doi:10.3390/electronics12183981_

Round 1

Reviewer 1 Report

The paper addresses the challenge of accurate target localization and grasping for NAO robots using monocular vision. It combines YOLOv8 network recognition with monocular ranging methods to improve accuracy. The paper introduces a visual distance error compensation model and a grasp control strategy based on pose interpolation. The key contributions of the paper include a monocular ranging model, visual distance error compensation, and multi-point measurement compensation technology.

The use of YOLOv8 for target recognition is promising, but the paper lacks details on the integration process. Providing more technical insights into adapting YOLOv8 to the NAO robot's environment would be beneficial.

Regarding the visual distance error compensation model effectively reduces errors to within 2 cm. However, further explanation of the model's mathematical foundation would enhance understanding. Also the grasp control strategy's effectiveness is evident, but more details about the algorithm and control mechanisms used for grasp adjustment would be valuable.

When referring to the experimental part of the research one can see that the experiments demonstrate the method's effectiveness, but more information on the setup, conditions, and datasets used would improve reproducibility and understanding.

In Conclusion the paper contributes to robotics by improving target localization and grasping for NAO robots. Enhancing technical details and insights into integration, model foundations, and experimental setups would benefit the paper. Nonetheless, the results demonstrate practical utility in achieving accurate object grasping by NAO robots, a significant achievement in robotics.

Reviewer 2 Report

Dear Authors,

1. Please select keywords from the following list.

2. Reference [10] not cited, please cite or remove.

3. Please paraphrase the "Introduction" section, because the similitude score is at 22%, which is still green, but it could be lower.

4. Figure 2 has the caption on the next page, please fix this.

5. Figure 10 (a) and (b) has the same caption, this should be corrected.

6. Before "Conclusions" it should be added also a "Discussions" section.

7. It should be added also an "Notations and Abbreviations" section before "References".

8. Please explain more clearly where do you used the YOLOv8 network, which are convolutional neural network models. While on the Figures 10 (b), 11 (a) (b) looks like it was used some image manipulation to track an object like: Gaussian Blur, convert RGB to HSV, a range setting mask, and after using erode and dilate masks. Finally it looks like it was done an edge detection. All this can be done without YOLOv8, where was this neural network used?

Best Regards

Reviewer 3 Report

The central part of the manuscript describes a monocular ranging model applied to objects (rods) recognised by a YOLOv8 network. There is also a section about a pose-interpolated grasping strategy, which appears unrelated to the other parts. The novelty of Section 4 is low.

Target recognition and localisation are well-performed and analysed. The method is tried in a field study with relevant experiments. There might be geometric configurations where the presented approach could fail, e.g., when the rod points away from the robot. A discussion about such cases is not present.

Figure 1: In the middle block, it seems that the term "Acquisition of internal and external" is not complete. A noun seems missing. Further, the diagram should be a bit enlarged for better readability.

The fonts in Figure 2 are too tiny. Please enlarge. Further, it is unclear what is shown.

ca. Line 75: ELAN, Neck, TAA, CIOU, DFL, and PAN are not introduced; these terms are only mentioned.

Line 95: The term O_i xy is unclear. Are xy indices?

Line 126, ff: Where do the values for a_1 ... a_5 come from?

Figure 6 is unclear. How were these values assessed?

Section 4: The role of Section 4 is unclear. What does this section contribute? Recommendation to remove this section or to explain how this supports recognition and localisation.

Figure 7: Please enlarge

Figure 9: Please enlarge. Which are the five joints? And which graph belongs to which joint of the NAO.

Figure 10: Are the pictures of (a) and (b) related to each other? It seems that one of these is mirrored. In the left image, there is a dark line which probably represents a rod where the edges are not visible. At first glance, I interpreted the area behind the rod as a step up to the blue sections. The thickness of the rod on the right and on the left are different.

Please note that the caption of Figure 10 b probably is not correct, as this is the processed image (not the original image).

Tables 1,2: Please make sure that these are two tables set aside one another.

Figure 12: Too tiny print. please use larger fonts and enlarge. Please improve the caption.

A comparison of the results presented in the manuscript with results from the literature is not present. How well do other methods perform compared to the authors'?

English language seems ok.

Reviewer 4 Report

The paper describes a method to detect a known object, such as a rod using YOLOv8 neural network and execute a monocular ranging algorithm to estimate distance in order to perform grasping activities with a NAO robot.

The paper is fairly written, altough some revision should be made, as some typos are still present and English can be improved troughout the paper.

Some issues or revisions are necessary:

- In the introduction, the state of the art can be improved by citing existing research such as monocular depth estimation using neural networks

- In line 74 it is stated that a "Mosaic data enhancement" is used. Maybe a citation is needed about this method?

- In the equation 1 probably some subscripts are missing (fx and fy)

- Line 127: It should probably clearly stated that these coefficients are computed based on your acquired dataset. Maybe a little specification of this should be made (i.e. how is the datased composed?)

- Figure 6a: add something like "measured distance" to the X (m) legend to improve readability

- Figure 10 (a,b): is the same viewpoint being represented? seems not. Change legend "(b) identified rod (or segmented image) captured by NAO robot (it is not an original image).

- Lines 203-209: Are you using the segmenation model of YOLOv8?

- Table 1 (and distance estimation in general): How can you precisely measure the distance value of the rod object? Maybe the diameter (size) of the rod is known a-priori?

- Line 60-61: Finally, *it is possible to* obtain the location coordinates ... , and *to* acquire the pose ...

- Figure 1: The top right box in the flow diagram has truncated text (Output target and ???)

- Line 83: represented -> represented below

- Line 83: possesses -> of

- Line 84: in the (X,Y) coordinate system -> in the image coordinate system

- Line 105-106: Similarly, it is possible to derive the position coordinates (Xw2, Yw2) of point Q2 under the robot's coordinate system.

- Line 113-115: Consequently, the deflection angle (e [keep your symbol]) of the target rod on the OwXwYw [please fix subscripts] plane can be optained, as demonstrated ...

- Line 127: Fix commas (seems not properly rendered on the downloaded pdf version)

Round 2

Reviewer 3 Report

The authors have addressed some of my comments. However, several editing-related comments have not been addressed sufficiently. For instance, Figure 2 could be enlarged for readability (use the entire paper width). Also, Figures 3, 4, and 5 should be enlarged for readability (also increase font size). Please enlarge Figure 8 (very tiny fonts).

Figure 10: Please describe which of the graphs belong to which part of the NAO robotic arm. Please enlarge (larger fonts).

Minor issue: Figures 11 and 12: The rod uses the entire width of the picture. This might be a bit confusing. In Figure 11a, it is unclear whether one sees the ends of the rod (or whether it continues potentially infinitely). Further, there is still the issue with the optical illusion that might occur, suggesting a step.

Line 321: I don't understand the text in this paragraph. Usually, this contains a list containing abbreviations and the term written out.

Section 4 still seems disjunct from the rest of the manuscript. Please add some lines about what each of the main contributions (Line 45ff) contributes beyond the current state of the art. Please motivate these goals by illustrating why these goals are needed.

It is further unclear what the NAO is going to grasp. Is it the rod? Please outline in the manuscript.

English language seems ok.
